# WORD2NET: DEEP REPRESENTATIONS OF LANGUAGE

## ABSTRACT

Word embeddings extract semantic features of words from large datasets of text. Most embedding methods rely on a log-bilinear model to predict the occurrence of a word in a context of other words. Here we propose *word2net*, a method that replaces their linear parametrization with neural networks. For each term in the vocabulary, word2net posits a neural network that takes the context as input and outputs a probability of occurrence. Further, word2net can use the hierarchical organization of its *word networks* to incorporate additional meta-data, such as syntactic features, into the embedding model. For example, we show how to share parameters across word networks to develop an embedding model that includes part-of-speech information. We study word2net with two datasets, a collection of Wikipedia articles and a corpus of U.S. Senate speeches. Quantitatively, we found that word2net outperforms popular embedding methods on predicting held-out words and that sharing parameters based on part of speech further boosts performance. Qualitatively, word2net learns interpretable semantic representations and, compared to vector-based methods, better incorporates syntactic information.

## 1 INTRODUCTION

Word embeddings are an important statistical tool for analyzing language, processing large datasets of text to learn meaningful vector representations of the vocabulary (Bengio et al., 2003; 2006; Mikolov et al., 2013b; Pennington et al., 2014). Word embeddings rely on the distributional hypothesis, that words used in the same contexts tend to have similar meanings (Harris, 1954). More informally (but equally accurate), a word is defined by the company it keeps (Firth, 1957).

While there are many extensions and variants of embeddings, most rely on a log-bilinear model. This model posits that each term is associated with an *embedding vector* and a *context vector*. Given a corpus of text, these vectors are fit to maximize an objective function that involves the inner product of each observed word's embedding with the sum of the context vectors of its surrounding words. With useful ways to handle large vocabularies, such as negative sampling (Mikolov et al., 2013a) or Bernoulli embeddings (Rudolph et al., 2016), the word embedding objective resembles a bank of coupled linear binary classifiers.

Here we introduce *word2net*, a word embedding method that relaxes this linear assumption. Word2net still posits a context vector for each term, but it replaces each word vector with a term-specific neural network. This *word network* takes in the sum of the surrounding context vectors and outputs the occurrence probability of the word. The word2net objective involves the output of each word's network evaluated with its surrounding words as input. The word2net objective resembles a bank of coupled *non-linear* binary classifiers.

How does word2net build on classical word embeddings? The main difference is that the word networks can capture non-linear interaction effects between co-occurring words; this leads to a better model of language. Furthermore, the word networks enable us to share per-term parameters based on word-level meta-data, such as syntactic information. Here we study word2net models that share parameters based on part-of-speech (POS) tags, where the parameters of certain layers of each network are shared by all terms tagged with the same POS tag.

Figure 1a illustrates the intuition behind word2net. Consider the term INCREASE. The top of the figure shows one observation of the word, i.e., one of the places in which it appears in the data. (This excerpt is from U.S. Senate speeches.) From this observation, the word2net objective contains the probability of a binary variable $w_{n,\text{INCREASE}}$ conditional on its context (i.e., the sum of the context vectors of the surrounding words). This variable is whether INCREASE occurred at position $n$.

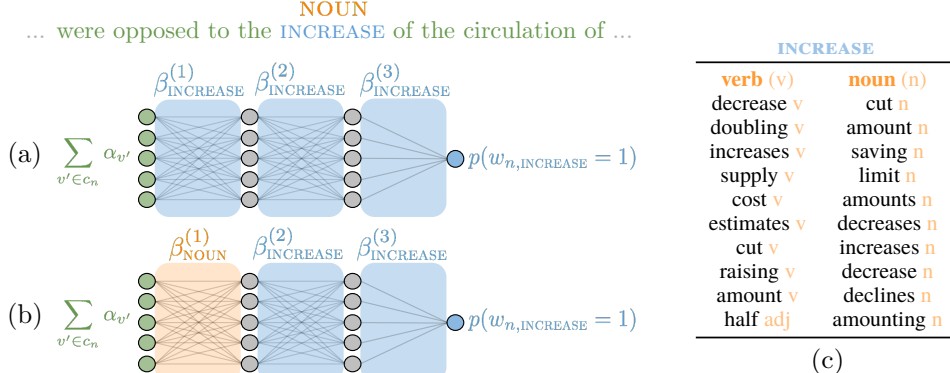

**Figure 1:** An illustration of word2net. **(a)** In word2net, each term $v$ is represented by a neural network with weights $\beta_v^{(\ell)}$. The word network predicts the probability of a target word (INCREASE, shown in blue) from its context (green). The input is the sum of the context vectors and the output is the occurrence probability of the target. **(b)** Word2net can incorporate syntactic information by sharing an entire layer (orange) between words with the same POS tag (NOUN in this case). **(c)** The fitted word2net with POS sharing can be queried for semantic similarities of the word networks for each word/tag pair. In this example, we list the most similar networks to INCREASE/VERB and INCREASE/NOUN.

The idea behind word2net is that the conditional probability of $w_{n,\text{INCREASE}}$ is the output of a multi-layer network that takes the context as input. Each layer of the network transforms the context into a new hidden representation, reweighting the latent features according to their relevance for predicting the occurrence of INCREASE. Note that not illustrated are the 0-variables, i.e., the negative samples, which correspond to words that are not at position $n$. In word2net, their probabilities also come from their corresponding word networks.

Now suppose we have tagged the corpus with POS. Figure 1b shows how to incorporate this syntactic information into word2net. The network is specific to INCREASE as a noun (as opposed to a verb). The parameters of the first layer (orange) are shared among all nouns in the collection; the other layers (blue) are specific to INCREASE. Thus, the networks for INCREASE/NOUN and INCREASE/VERB differ in how the first layer promotes the latent aspects of the context, i.e., according to which context features are more relevant for each POS tag. This model further lets us consider these two POS tags separately. Figure 1c shows the most similar words to each sense of INCREASE; the method correctly picks out tagged words related to the verb and related to the noun.

Below, we develop the details of word2net and study its performance with two datasets, a collection of Wikipedia articles and a corpus of U.S. Senate speeches. We found that word2net outperforms popular embedding methods on predicting held-out words, and that sharing parameters based on POS further boosts performance. Qualitatively, word2net learns interpretable semantic representations and, compared to vector-based methods, better incorporates syntactic information.

**Related work.** Word2net builds on word embeddings methods. Though originally designed as deep neural network architectures (Bengio et al., 2003; 2006; Mnih & Hinton, 2007), most applications of word embeddings now rely on log-bilinear models (Mikolov et al., 2013a;b;c; Pennington et al., 2014; Mnih & Teh, 2012; Mnih & Kavukcuoglu, 2013; Levy & Goldberg, 2014; Vilnis & McCallum, 2015; Barkan, 2016; Bamler & Mandt, 2017). The key innovation behind word2net is that it represents words with functions, instead of vectors (Rumelhart et al., 1986) or distributions (Vilnis & McCallum, 2015). Word2net keeps context vectors, but it replaces the embedding vector with a neural network.

Previous work has also used deep neural networks for word embeddings (Bengio et al., 2003; 2006; Mnih & Hinton, 2007); these methods use a single network that outputs the unnormalized log probabilities for all words in the vocabulary. Word2net takes a different strategy: it has a separate network for each vocabulary word. Unlike the previous methods, word2net's approach helps maintain the objective as a bank of binary classifiers, which allows for faster optimization of the networks.

To develop word2net, we adopt the perspective of exponential family embeddings (Rudolph et al., 2016), which extend word embeddings to data beyond text. There are several extensions to exponential

family embeddings (Rudolph & Blei, 2017; Rudolph et al., 2017; Liu & Blei, 2017), but they all have in common an exponential family likelihood whose natural parameter has a log-bilinear form. Word2net extends this framework to allow for non-linear relationships. Here we focus on Bernoulli embeddings, which are related to word embeddings with negative sampling, but our approach easily generalizes to other exponential family distributions (e.g., Poisson).

Finally, word embeddings can capture semantic properties of the word, but they tend to neglect most of the syntactic information (Andreas & Klein, 2014). Word2net introduces a simple way to leverage the syntactic information to improve the quality of the word representations.

## 2 WORD2NET

In this section we develop word2net as a novel extension of Bernoulli embeddings (Rudolph et al., 2016). Bernoulli embeddings are a conditional model of text, closely related to word2vec. Specifically, they are related to continuous bag-of-words (CBOW) with negative sampling.[1] We first review Bernoulli embeddings and then we present word2net as a deep Bernoulli embedding model.

### 2.1 BACKGROUND: BERNOULLI EMBEDDINGS

Exponential family embeddings learn an embedding vector $\rho_v \in \mathbb{R}^K$ and a context vector $\alpha_v \in \mathbb{R}^K$ for each unique term in the vocabulary, $v = 1, \ldots, V$. These vectors encode the semantic properties of words, and they are used to parameterize the conditional probability of a word given its context. Specifically, let $w_n$ be the $V$-length one-hot vector indicating the word at location $n$, such that $w_{nv} = 1$ for one term (vocabulary word) $v$, and let $c_n$ be the indices of the words in a fixed-sized window centered at location $n$ (i.e., the indices of the context words). Exponential family embeddings parameterize the conditional probability of the target word given its context via a linear combination of the embedding vector and the context vectors,

$$p(w_{nv} \mid c_n) = \text{Bernoulli}\big(\sigma(\rho_v^\top \Sigma_n)\big), \quad \text{with} \quad \Sigma_n \triangleq \sum_{v' \in c_n} \alpha_{v'}. \tag{1}$$

Here, $\sigma(x) = \frac{1}{1+e^{-x}}$ is the sigmoid function, and we have introduced the notation $\Sigma_n$ for the sum of the context vectors at location $n$. Note that Eq. 1 does not impose the constraint that the sum over the vocabulary words $\sum_v p(w_{nv} = 1 \mid c_n)$ must be 1. This significantly alleviates the computational complexity (Mikolov et al., 2013b; Rudolph et al., 2016).

This type of exponential family embedding is called Bernoulli embedding, named for its conditional distribution. In Bernoulli embeddings, our goal is to learn the embedding vectors $\rho_v$ and the context vectors $\alpha_v$ from the text by maximizing the log probability of words given their contexts. The data contains $N$ pairs $(w_n, c_n)$ of words and their contexts, and thus we can form the objective function $\mathcal{L}(\rho, \alpha)$ as the sum of $\log p(w_{nv} \mid c_n)$ for all instances and vocabulary words. The resulting objective can be seen as a bank of $V$ binary classifiers, where $V$ is the vocabulary size. To see that, we make use of Eq. 1 and express the objective $\mathcal{L}(\rho, \alpha)$ as a sum over vocabulary words,

$$\mathcal{L}(\rho, \alpha) = \sum_{n=1}^N \sum_{v=1}^V \log p(w_{nv} \mid c_n) = \sum_{v=1}^V \left( \sum_{n: w_{nv}=1} \log \sigma(\rho_v^\top \Sigma_n) + \sum_{n: w_{nv}=0} \log \sigma(-\rho_v^\top \Sigma_n) \right). \tag{2}$$

If we hold all the context vectors $\alpha_v$ fixed, then Eq. 2 is the objective of $V$ independent logistic regressors, each predicting whether a word appears in a given context or it does not. The positive examples are those where word $v$ actually appeared in a given context; the negative examples are those where $v$ did not appear. It is the context vectors that couple the $V$ binary classifiers together.

In practice, we need to either downweight the contribution of the zeros in Eq. 2, or subsample the set of negative examples for each $n$ (Rudolph et al., 2016). We follow the latter case here, which leads to negative sampling (Mikolov et al., 2013b). (See the connection in more detail in Appendix B.)

---

[1]See Appendix B for more details on the connections.

## 2.2 Word2Net as a deep Bernoulli embedding model

Word2net replaces the linear classifiers in Eq. 2 with non-linear classifiers. In particular, we replace the linear combination $\rho_v^\top \Sigma_n$ with a neural network that is specific to each vocabulary word $v$, so that

$$p(w_{nv} = 1 \mid c_n) = \sigma\big(f\left(\Sigma_n; \ \beta_v\right)\big), \tag{3}$$

where $f(\cdot \ ; \ \beta_v) : \mathbb{R}^K \to \mathbb{R}$ is a feed-forward neural network with parameters (i.e., weights and intercepts) $\beta_v$. The number of neurons of the input layer is $K$, equal to the length of the context vectors $\alpha_v$. Essentially, we have replaced the per-term embedding vectors $\rho_v$ with a per-term neural network $\beta_v$. We refer to the per-term neural networks as *word networks*.

The word2net objective is the sum of the log conditionals,

$$\mathcal{L}_{\text{word2net}}(\rho, \alpha) = \sum_{v=1}^{V} \left( \sum_{n: \, w_{nv}=1} \log \sigma\big(f\left(\Sigma_n; \ \beta_v\right)\big) + \sum_{n: \, w_{nv}=0} \log \sigma\big(-f\left(\Sigma_n; \ \beta_v\right)\big) \right), \tag{4}$$

where we choose the function $f(\cdot \ ; \ \beta_v)$ to be a three-layer neural network,[2]

$$h_{nv}^{(1)} = \tanh\left(\Sigma_n^\top \beta_v^{(1)}\right), \quad h_{nv}^{(2)} = \tanh\left((h_{nv}^{(1)})^\top \beta_v^{(2)}\right), \quad f\left(\Sigma_n; \ \beta_v\right) = (h_{nv}^{(2)})^\top \beta_v^{(3)}. \tag{5}$$

Replacing vectors with neural networks has several implications. First, the bank of binary classifiers has additional model capacity to capture nonlinear relationships between the context and the co-occurrence probabilities. Specifically, each layer consecutively transforms the context to a different representation until the weight matrix at the last layer can linearly separate the real occurrences of the target word from the negative examples.

Second, for a fixed dimensionality $K$, the resulting model has more parameters.[3] This increases the model capacity, but it also increases the risk of overfitting. Indeed, we found that without extra regularization, the neural networks may easily overfit to the training data. We regularize the networks via either weight decay or parameter sharing (see below). In the empirical study of Section 3 we show that word2net fits text data better than its shallow counterparts and that it captures semantic similarities. Even for infrequent words, the learned semantic representations are meaningful.

Third, we can exploit the hierarchical structure of the neural network representations via parameter sharing. Specifically, we can share the parameters of a specific layer of the networks of different words. This allows us to explicitly account for POS tags in our model (see below).

**Regularization through parameter sharing enables the use of POS tags.** One way to regularize word2net is through parameter sharing. For parameter sharing, each word is assigned to one of $T$ groups. Importantly, different occurrences of a term may be associated to different groups.

We share specific layers of the word networks among words in the same group. In this paper, all neural network representations have 3 layers. We use index $\ell \in \{1, 2, 3\}$ to denote the layer at which we apply the parameter sharing. Then, for each occurrence of term $v$ in group $t$ we set $\beta_v^{(\ell)} = \beta_t^{(\ell)}$.

Consider now two extreme cases. First, for $T = 1$ group, we have a strong form of regularization by forcing all word networks to share the parameters of layer $\ell$. The number of parameters for layer $\ell$ has been divided by the vocabulary size, which implies a reduction in model complexity that might help prevent overfitting. This parameter sharing structure does not require side information and hence can be applied to any text corpus. In the second extreme case, each word is in its own group and $T = V$. This set-up recovers the model of Eqs. 4 and 5, which does not have parameter sharing.

When we have access to a corpus annotated with POS tags, parameter sharing lets us use the POS information to improve the capability of word2net by capturing the semantic structure of the data. Andreas & Klein (2014) have shown that word embeddings do not necessarily encode much syntactic information, and it is still unclear how to use syntactic information to learn better word embeddings. The main issue is that many words can appear with different tags; for example, FISH can be both a NOUN and refer to the animal or a VERB and refer to the activity of catching the animal. On the one hand, both meanings are related. On the other hand, they may have differing profiles of which

---

[2] Three layers performed well in our experiments, allowing for parameter sharing to include POS tags.
[3] For fairness, in Section 3 we also compare to shallow models with the same number of parameters.

contexts they appear in. Ideally, embedding models should be able to capture the difference. However, the simple approach of considering FISH/NOUN and FISH/VERB as separate terms fails because there are few occurrences of each individual term/tag pair. (We show that empirically in Section 3.)

Exploiting the hierarchical nature of the network representations of word2net, we incorporate POS information through parameter sharing as follows. Assume that for location $n$ in the text we have a one-hot vector $s_n \in \{0, 1\}^T$ indicating the POS tag. To model the observation at position $n$, we use a neural network specific to that term/tag combination,

$$p(w_{nv} = 1, s_{nt} = 1 \mid c_n) = \sigma \left( f \left( \Sigma_n;\ \beta_v^{(\neg \ell)}, \beta_t^{(\ell)} \right) \right). \tag{6}$$

That is, the neural network parameters are combined to form a neural network in which layer $\ell$ has parameters $\beta_t^{(\ell)}$ and the other layers have parameters $\beta_v^{(\neg \ell)}$. Thus, we leverage the information about the POS tag $t$ by replacing $\beta_v^{(\ell)}$ with $\beta_t^{(\ell)}$ in layer $\ell$, resulting in POS parameter sharing at that layer. If the same term $v$ appears at a different position $n'$ with a different POS tag $t'$, at location $n'$ we replace the parameters $\beta_v^{(\ell)}$ of layer $\ell$ with $\beta_{t'}^{(\ell)}$. Figure 1b illustrates POS parameter sharing at $\ell = 1$.

Even though now we have a function $f(\cdot)$ for each term/tag pair, the number of parameters does *not* scale with the product $V \times T$; indeed the number of parameters of the network with POS information is smaller than the number of parameters of the network without side information (Eq. 5). The reason is that the number of parameters necessary to describe one of the layers has been reduced from $V$ to $T$ due to parameter sharing (the other layers remain unchanged).

Finally, note that we have some flexibility in choosing which layer is tag-specific and which layers are word-specific. We explore different combinations in Section 3, where we show that word2net with POS information improves the performance of word2net. The parameter sharing approach extends to side information beyond POS tags, as long as the words can be divided into groups, but we focus on parameter sharing across all words ($T = 1$) or across POS tags.

**Semantic similarity of word networks.** In standard word embeddings, the default choice to compute semantic similarities between words is by cosine distances between the word vectors. Since word2net replaces the word vectors with word networks, we can no longer apply this default choice. We next describe the procedure that we use to compute semantic similarities between word networks.

After fitting word2net, each word is represented by a neural network. Given that these networks parameterize functions, we design a metric that accounts for the fact that two functions are similar if they map similar inputs to similar outputs. So the intuition behind our procedure is as follows: we consider a set of $K$-dimensional inputs, we evaluate the output of each neural network on this set of inputs, and then we compare the outputs across networks. For the inputs, we choose the $V$ context vectors, which we stack together into a matrix $\boldsymbol{\alpha} \in \mathbb{R}^{V \times K}$. We evaluate each network $f(\cdot)$ row-wise on $\boldsymbol{\alpha}$ (i.e., feeding each $\alpha_v$ as a $K$-dimensional input to obtain a scalar output), obtaining a $V$-dimensional summary of where the network $f(\cdot)$ maps the inputs. Finally, we use the cosine distance of the outputs to compare the outputs across networks. In summary, we obtain the similarity of two words $w$ and $v$ as

$$\text{dist}(w, v) = \frac{f(\boldsymbol{\alpha};\ \beta_w)^\top f(\boldsymbol{\alpha};\ \beta_v)}{||f(\boldsymbol{\alpha};\ \beta_w)||_2\ ||f(\boldsymbol{\alpha};\ \beta_v)||_2}. \tag{7}$$

If we are using parameter sharing, we can also compare POS-tagged words; e.g., we may ask how similar is FISH/NOUN to FISH/VERB. The two combinations will have different representations under the word2net method trained with POS-tag sharing. Assuming that layer $\ell$ is the shared layer, we compute the semantic similarity between the word/tag pair $[w, t]$ and the pair $[v, s]$ as

$$\text{dist}([w, t], [v, s]) = \frac{f(\boldsymbol{\alpha};\ \beta_w^{(\neg \ell)}, \beta_t^{(\ell)})^\top f(\boldsymbol{\alpha};\ \beta_v^{(\neg \ell)}, \beta_s^{(\ell)})}{||f(\boldsymbol{\alpha};\ \beta_w^{(\neg \ell)}, \beta_t^{(\ell)})||_2\ ||f(\boldsymbol{\alpha};\ \beta_v^{(\neg \ell)}, \beta_s^{(\ell)})||_2}. \tag{8}$$

## 3 EMPIRICAL RESULTS

In this section we study the performance of word2net on two datasets, Wikipedia articles and Senate speeches. We show that word2net fits held-out data better than existing models and that the learned network representations capture semantic similarities. Our results also show that word2net is superior

**Table 1:** Summary of the two corpora analyzed in Section 3.

|  | corpus size | vocabulary | tagger | tags | tagged vocabulary |
|---|---|---|---|---|---|
| **Wikipedia** | 17M words | 15K terms | NLTK | 11 tags | 49K tagged terms |
| **Senate speeches** | 24M words | 15K terms | CoreNLP | 11 tags | 38K tagged terms |

at incorporating syntactic information into the model, which improves both the predictions and the quality of the word representations.

**Data.** We use word2net to study two data sets, both with and without POS tags:

*Wikipedia:* The text8 corpus is a collection of Wikipedia articles, containing 17M words. We form a vocabulary with the 15K most common terms, replacing less frequent terms with the UNKNOWN token. We annotate text8 using the NLTK POS tagger and the universal tagset.[4] Table 7 in Appendix C shows a description of the tagset. We also form a tagged dataset in which each term/tag combination has a unique token, resulting in a vocabulary of 49K tagged terms.

*Senate speeches:* These are the speeches given in the U.S. Senate in the years 1916-2009. The data is a transcript of spoken language and contains 24M words. Similarly as above, we form a vocabulary of 15K terms. We annotate the text using the Stanford CoreNLP POS tagger (Manning et al., 2014), and we map the tags to the universal tagset. We form a tagged dataset with 38K tagged terms.

Table 1 summarizes the information about both corpora. We split each dataset into a training, a validation, and a test set, which respectively contain 90%, 5%, and 5% of the words. Additional details on preprocessing are in Appendix C.

**Methods.** We compare word2net to its shallow counterpart, the CBOW model (Mikolov et al., 2013b), which is equivalent to Bernoulli embeddings (B-EMB)[5] (Rudolph et al., 2016). We also compare with the skip-gram model.[6] (Mikolov et al., 2013b) We run B-EMB/CBOW and skip-gram on the data and also on the augmented data of POS-tagged terms. In detail, the methods we compare are:

- B-EMB/CBOW: Learns vector representations for each word (or tagged word) by optimizing Eq. 2.
- *Skip-gram*: Learns vector representations for each word (or tagged word) by optimizing Eq. 12.
- *Word2net*: Learns a neural network representation for each word by optimizing Eq. 4. We study the following parameter sharing schemes:
    1. ⬤⬤⬤ : no parameter sharing.
    2. ◯ ALL ◯ : layer $\ell$ shared between all networks.
    3. ◯ POS ◯ : layer $\ell$ shared between terms with the same part-of-speech (POS) tag.

For word2net, we experiment with the context dimensions $K \in \{20, 100\}$. The context dimension is also the dimension of the input layer. For $K = 20$, we use $H_1 = 10$ hidden units in the first hidden layer of each word network and $H_2 = 10$ hidden units in the second layer. For $K = 100$, we use $H_1 = H_2 = 20$ hidden units. Without parameter sharing, the number of parameters per word is $K + KH_1 + H_1H_2 + H_2$. The shallow models have $2K$ parameters per term (the entries of the context and word vectors). Since we want to compare models both in terms of context dimension $K$ and in terms of total parameters, we fit the methods with $K \in \{20, 165, 100, 1260\}$.

We experiment with context sizes $|c_n| \in \{2, 4, 8\}$ and we train all methods using stochastic gradient descent (SGD) (Robbins & Monro, 1951) with $|\mathcal{S}_n| = 10$ negative samples on the Wikipedia data and with $|\mathcal{S}_n| = 20$ negative samples on the Senate speeches. We use L2 regularization with standard deviation 10 for the word and context vectors, as well as weight decay for the neural networks. We use Adam (Kingma & Ba, 2015) with Tensorflow's default settings (Abadi et al., 2016) to train all methods for up to 30000 iterations, using a minibatch size of 4069 or 1024. We assess convergence by monitoring the loss on a held-out validation set every 50 iterations, and we stop training when the average validation loss starts increasing. We initialize and freeze the context vectors of the word2net methods with the context vectors from a pretrained Bernoulli embedding with the same context dimension $K$. Network parameters are initialized according to standard initialization schemes of

---

[4]See http://nltk.org.

[5]See Appendix B for the detailed relationship between B-EMB and CBOW with negative sampling.

[6]The skip-gram objective is related to CBOW/B-EMB through Jensen's inequality (see Appendix B).

**Table 2:** Word2net outperforms existing word embedding models (skip-gram and B-EMB/CBOW) in terms of test log-likelihood on the Wikipedia data, both with and without POS tags. We compare models with the same context dimension $K$ and the same total number of parameters $p/V$ for different context sizes (cs). (Results on more configurations are in Appendix A.) For word2net, we study different parameter sharing schemes, and the color coding indicates which layer is shared and how, as in Figure 1. Parameter sharing improves the performance of word2net, especially with POS tags.

| | vocabulary | $K$ | $p/V$ | cs 2 | cs 4 | cs 8 |
|---|---|---|---|---|---|---|
| Mikolov et al. (2013b): | | | | | | |
| skip-gram | words | 20 | 40 | −1.061 | −1.062 | −1.071 |
| skip-gram | tagged words | 20 | 240 | −2.994 | −3.042 | −3.042 |
| Mikolov et al. (2013b); Rudolph et al. (2016): | | | | | | |
| B-EMB/CBOW | words | 20 | 40 | −1.023 | −0.976 | −0.941 |
| B-EMB/CBOW | words | 165 | 330 | −1.432 | −1.388 | −1.381 |
| B-EMB/CBOW | tagged words | 20 | 240 | −1.411 | −1.437 | −1.461 |
| **this work:** | sharing | | | | | |
| word2net | (none) | 20 | 330 | −0.940 | −0.912 | −0.937 |
| word2net | ALL (layer 1) | 20 | ≈ 120 | −1.040 | −1.003 | −0.964 |
| word2net | ALL (layer 2) | 20 | ≈ 230 | −1.191 | −1.141 | −1.111 |
| word2net | ALL (layer 3) | 20 | ≈ 320 | −0.863 | −0.881 | −0.890 |
| word2net | POS (layer 1) | 20 | ≈ 120 | −0.918 | −0.914 | −0.871 |
| word2net | POS (layer 2) | 20 | ≈ 230 | −0.844 | **−0.801** | **−0.793** |
| word2net | POS (layer 3) | 20 | ≈ 320 | **−0.840** | −0.822 | −0.862 |

feed-forward neural networks (Glorot & Bengio, 2010), i.e., the weights are initialized from a uniform distribution with bounds $\pm\sqrt{6}/\sqrt{H_{\text{in}} + H_{\text{out}}}$.

**Quantitative results: Word2net has better predictive performance.** We compute the predictive log-likelihood of the words in the test set, $\log p(w_{nv} \mid c_n)$. For skip-gram, which was trained to predict the context words from the target, we average the context vectors $\alpha_v$ for a fair comparison.[7]

Table 2 shows the results for the Wikipedia dataset. We explore different model sizes: with the same number of parameters as word2net, and with the same dimensionality $K$ of the context vectors. For word2net, we explore different parameter sharing approaches. Table 5 in Appendix A shows the results for other model sizes (including $K = 100$). In both tables, word2net without parameter sharing performs at least as good as the shallow models. Importantly, the performance of word2net improves with parameters sharing, and it outperforms the other methods.

Tables 2 and 5 also show that B-EMB/CBOW and skip-gram perform poorly when we incorporate POS information by considering an augmented vocabulary of tagged words. The reason is that each term becomes less frequent, and these approaches would require more data to capture the co-occurrence patterns of tagged words. In contrast, word2net with POS parameter sharing provides the best predictions across all methods (including other versions of word2net).

Finally, Table 6 in Appendix A shows the predictive performance for the U.S. Senate speeches. On this corpus, skip-gram performs better than B-EMB/CBOW and word2net without parameter sharing; however, word2net with POS sharing also provides the best predictions across all methods.

**Qualitative results: Word2net captures similarities and leverages syntactic information.** Table 3 displays the similarity between word networks (trained on Wikipedia with parameter sharing at layer $\ell = 1$), compared to the similarities captured by word embeddings (B-EMB/CBOW). For each query word, we list the three most similar terms, according to the learned representations. The word vectors are compared using cosine similarity, while the word networks are compared using Eq. 7. The table shows that word2net can capture latent semantics, even for less frequent words such as PARROT.

Table 4 shows similarities of models trained on the Senate speeches. In particular, the table compares: B-EMB/CBOW without POS information, B-EMB/CBOW trained on the augmented vocabulary of tagged words, and word2net with POS parameter sharing at the input layer ($\ell = 1$). We use Eq. 8 to compute the similarity across word networks with POS sharing. We can see that word2net is superior at incorporating syntactic information into the learned representations. For example, the most similar

---

[7]If we do not average, the held-out likelihood of skip-gram becomes worse.

**Table 3:** The word networks fitted using word2net capture semantic similarities. We compare the top 3 similar words to several query words (shaded in gray) for CBOW/B-EMB and word2net, trained on the Wikipedia dataset. The numbers in parenthesis indicate the frequency of the query words.

| RATE (3000) | | VOTE (1000) | | COFFEE (500) | | PARROT (70) | |
|---|---|---|---|---|---|---|---|
| **CBOW** | **word2net** | **CBOW** | **word2net** | **CBOW** | **word2net** | **CBOW** | **word2net** |
| expectancy | capacity | votes | elect | bananas | beans | turtle | dolphin |
| per | amount | voting | candidate | potatoes | juice | beaver | crow |
| increase | energy | election | candidates | pepper | poultry | pratchett | dodo |

**Table 4:** Word2net learns better semantic representations by exploiting syntactic information. The top 3 similar words to several queries are listed for different models fitted to the Senate speeches. We compare CBOW trained without POS tags (left), CBOW with POS tags (center), and word2net with POS parameter sharing (right). The POS tags are noted in orange. Parameter sharing helps word2net capture better semantic similarities, while adding the POS information to CBOW hurts its performance.

| ME (pron) | | | BECAUSE (sc) | | |
|---|---|---|---|---|---|
| **CBOW** | **CBOW POS** | **word2net** | **CBOW** | **CBOW POS** | **word2net** |
| like | governor n | myself pron | but | unemployed n | as sc |
| senator | senator from alabama n | my pron | reason | annuity n | that sc |
| just | used adj | himself pron | that | shelled v | through sc |

| CAUSES (n) | | | SAY (v) | | |
|---|---|---|---|---|---|
| **CBOW** | **CBOW POS** | **word2net** | **CBOW** | **CBOW POS** | **word2net** |
| fatal | pro adj | consequences n | think | time v | think v |
| consequences | enough adv | clash n | what | best adj | know v |
| coupled | positions n | handicaps n | just | favour n | answer v |

networks to the pronoun ME are other pronouns such as MYSELF, MY, and HIMSELF. Word networks are often similar to other word networks with the same POS tag, but we also see some variation. One such example is in Figure 1c, which shows that the list of the 10 most similar words to the verb INCREASE contains the adjective HALF.

## 4 DISCUSSION

We have presented word2net, a method for learning neural network representations of words. The word networks are used to predict the occurrence of words in small context windows and improve prediction accuracy over existing log-bilinear models. We combine the context vectors additively, but this opens the door for future research directions in which we explore other ways of combining the context information, such as accounting for the order of the context words and their POS tags.

We have also introduced parameter sharing as a way to share statistical strength across groups of words and we have shown empirically that it improves the performance of word2net. Another opportunity for future work is to explore other types of parameter sharing besides POS sharing, such as sharing layers across documents or learning a latent group structure together with the word networks.

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

APPENDIX

## A   ADDITIONAL RESULTS

For completeness, we show here some additional results that we did not include in the main text for space constraints.

In particular, Table 5 compares the test log-likelihood of word2net with the competing models—namely, skip-gram and B-EMB/CBOW. All methods are trained with negative sampling, as described in the main text. This table shows the results for the Wikipedia dataset, similarly to Table 2, but it includes other model sizes (i.e., another value of $K$). In this table, word2net with no parameter sharing performs similarly to B-EMB/CBOW with the same number of parameters, but its performance can be further improved with part-of-speech (POS) parameter sharing.

Table 6 shows the test log-likelihood for the U.S. Senate speeches. Here, skip-gram is the best method that does not use POS tags, but it is outperformed by word2net with POS parameter sharing.

**Table 5:** Comparison of the test log-likelihood across different models on the Wikipedia dataset. We compare models with the same context dimension $K$ and the same total number of parameters $p/V$ for different context sizes ("cs"). For word2net, we explore different parameter sharing schemes. The color coding of the parameter sharing (same as Figure 1) indicates which layer is shared and how.

| | vocabulary | $K$ | $p/V$ | cs 2 | cs 4 | cs 8 |
|---|---|---|---|---|---|---|
| Mikolov et al. (2013b): | | | | | | |
| skip-gram | words | 100 | 200 | $-1.107$ | $-1.053$ | $-1.043$ |
| skip-gram | tagged words | 100 | 1200 | $-3.160$ | $-3.151$ | $-3.203$ |
| Mikolov et al. (2013b); Rudolph et al. (2016): | | | | | | |
| B-EMB/CBOW | words | 100 | 200 | $-1.212$ | $-1.160$ | $-1.127$ |
| B-EMB/CBOW | tagged words | 100 | 1200 | $-1.453$ | $-3.387$ | $-3.433$ |
| B-EMB/CBOW | words | 1260 | 2520 | $-3.772$ | $-2.397$ | $-2.506$ |
| **this work:** | sharing | | | | | |
| word2net | ⬭⬭⬭ | 100 | 2520 | $-1.088$ | $-1.049$ | $-1.012$ |
| word2net | ALL ⬭⬭ | 100 | $\approx 520$ | $-1.041$ | $-0.988$ | $-1.001$ |
| word2net | ⬭ ALL ⬭ | 100 | $\approx 2120$ | $-1.114$ | $-1.059$ | $-1.016$ |
| word2net | POS ⬭⬭ | 100 | $\approx 521$ | $\mathbf{-0.828}$ | $\mathbf{-0.807}$ | $\mathbf{-0.770}$ |
| word2net | ⬭ POS ⬭ | 100 | $\approx 2120$ | $-0.892$ | $-0.850$ | $-0.822$ |

## B   RELATION BETWEEN BERNOULLI EMBEDDINGS AND WORD2VEC

Word2vec (Mikolov et al., 2013b) is one of the most widely used method for learning vector representations of words. There are multiple ways to implement word2vec. First, there is a choice of the objective. Second, there are several ways of how to approximate the objective to get a scalable algorithm. In this section, we describe the two objectives, continuous bag-of-words (CBOW) and skip-gram, and we focus on negative sampling as the method of choice to achieve scalability. We describe the similarities and differences between Bernoulli embeddings (Rudolph et al., 2016) and these two objectives. In summary, under certain assumptions Bernoulli embeddings are equivalent to CBOW with negative sampling, and are related to skip-gram through Jensen's inequality.

> B-EMB $\equiv$ CBOW (negative sampling)

First we explain how Bernoulli embeddings and CBOW with negative sampling are related. Consider the Bernoulli embedding full objective,

$$\mathcal{L}(\rho, \alpha) = \sum_n \left( \sum_{v:\, w_{nv}=1} \log \sigma(\rho_v^\top \Sigma_n) + \sum_{v:\, w_{nv}=0} \log \sigma(-\rho_v^\top \Sigma_n) \right). \tag{9}$$

In most cases, the summation over negative examples ($w_{nv} = 0$) is computationally expensive to compute. To address that, we form an unbiased estimate of that term by subsampling a random set $\mathcal{S}_n$

**Table 6:** Comparison of the test log-likelihood across different models on the Senate speeches. We compare models with the same context dimension $K$ and the same total number of parameters $p/V$ for different context sizes ("cs"). For word2net, we explore different parameter sharing schemes. The color coding of the parameter sharing (same as Figure 1) indicates which layer is shared and how.

| | vocabulary | $K$ | $p/V$ | cs 2 | cs 4 | cs 8 |
|---|---|---|---|---|---|---|
| Mikolov et al. (2013b): | | | | | | |
| skip-gram | words | 20 | 40 | $-1.052$ | $-1.080$ | $-1.061$ |
| skip-gram | tagged words | 20 | 240 | $-1.175$ | $-1.199$ | $-1.227$ |
| Mikolov et al. (2013b); Rudolph et al. (2016): | | | | | | |
| B-EMB/CBOW | words | 20 | 40 | $-1.274$ | $-1.246$ | $-1.222$ |
| B-EMB/CBOW | tagged words | 20 | 240 | $-1.352$ | $-1.340$ | $-1.339$ |
| B-EMB/CBOW | words | 165 | 330 | $-1.735$ | $-1.734$ | $-1.744$ |
| **this work:** | sharing | | | | | |
| word2net | | 20 | 330 | $-1.406$ | $-1.555$ | $-1.401$ |
| word2net | ALL | 20 | $\approx 120$ | $-1.276$ | $-1.256$ | $-1.243$ |
| word2net | ALL | 20 | $\approx 230$ | $-1.462$ | $-1.435$ | $-1.413$ |
| word2net | POS | 20 | $\approx 120$ | $\mathbf{-0.873}$ | $\mathbf{-0.860}$ | $\mathbf{-0.850}$ |
| word2net | POS | 20 | $\approx 230$ | $-1.057$ | $-1.034$ | $-1.015$ |

of terms and rescaling by $\frac{V-1}{|\mathcal{S}_n|}$,

$$\widehat{\mathcal{L}}(\rho, \alpha) = \sum_n \left( \sum_{v:\, w_{nv}=1} \log \sigma(\rho_v^\top \Sigma_n) + \gamma \frac{V-1}{|\mathcal{S}_n|} \sum_{v \in \mathcal{S}_n} \log \sigma(-\rho_v^\top \Sigma_n) \right). \qquad (10)$$

Here, we have introduced an auxiliary coefficient $\gamma$. The estimate is unbiased only for $\gamma = 1$; however, Rudolph et al. (2016) showed that downweighting the contribution of the zeros works better in practice.[8] In particular, if we set the downweight factor as $\gamma = \frac{|\mathcal{S}_n|}{V-1}$, we recover the objective of CBOW with negative sampling,

$$\widehat{\mathcal{L}}(\rho, \alpha) = \sum_n \left( \sum_{v:\, w_{nv}=1} \log \sigma(\rho_v^\top \Sigma_n) + \sum_{v \in \mathcal{S}_n} \log \sigma(-\rho_v^\top \Sigma_n) \right) \equiv \mathcal{L}_{\text{CBOW}}(\rho, \alpha) \qquad (11)$$

There are two more subtle theoretical differences between both. The first difference is that Bernoulli embeddings include a regularization term for the embedding vectors, whereas CBOW does not. The second difference is that, in Bernoulli embeddings, we need to draw a new set of negative samples $\mathcal{S}_n$ at each iteration of the gradient ascent algorithm (because we form a noisy estimator of the downweighted objective). In contrast, in CBOW with negative sampling, the samples $\mathcal{S}_n$ are drawn once in advance and then hold fixed. In practice, for large datasets, we have not observed significant differences in the performance of both approaches. For simplicity, we draw the negative samples $\mathcal{S}_n$ only once.

> CBOW (negative sampling) $\geq$ skip-gram (negative sampling)

Now we show how CBOW and skip-gram are related (considering negative sampling for both). Recall that the objective of CBOW is to predict a target word from its context, while the skip-gram objective is to predict the context from the target word. Negative sampling breaks the multi-class constraint that the sum of the probability of each word must equal one, and instead models probabilities of the individual entries of the one-hot vectors representing the words.

When we apply negative sampling, the CBOW objective becomes Eq. 11. The skip-gram objective is given by

$$\mathcal{L}_{\text{skip-gram}}(\rho, \alpha) = \sum_{(n,v):\, w_{nv}=1} \left( \sum_{v' \in c_n} \log \sigma\left(\rho_v^\top \alpha_{v'}\right) + \sum_{v' \in \mathcal{S}_n} \log \sigma\left(-\rho_v^\top \alpha_{v'}\right) \right), \qquad (12)$$

---

[8]This is consistent with the approaches in recommender systems (Hu et al., 2008).

**Table 7:** Universal POS tagset.

| Tag | Description |
|-----|-------------|
| adj | adjective |
| adp | adposition |
| adv | adverb |
| conj | conjuction |
| det | determiner, article |
| n | noun |
| num | numeral |
| prt | particle |
| pron | pronoun |
| sc | preposition or subordinating conjuction |
| v | verb |
| x | other |

That is, for each target term $w_{nv}$, the CBOW objective has one term while the skip-gram objective has $|c_n|$ terms. Consider a term $(n, v)$ for which $w_{nv} = 1$. We take the corresponding CBOW term from Eq. 11 and we apply Jensen's inequality to obtain the corresponding skip-gram term in Eq. 12:

$$\log \sigma(\rho_v^\top \Sigma_n) = \log \sigma \left( \rho_v^\top \sum_{v' \in c_n} \alpha_{v'} \right) \geq \sum_{v' \in c_n} \log \sigma \left( \rho_v^\top \alpha_{v'} \right). \tag{13}$$

Here, we have made use of the concavity of the $\log \sigma(\cdot)$ function. In general, this is a consequence of the convexity of the log-normalizer of the (Bernoulli) exponential family distribution.

This holds for the "positive" examples $w_{nv}$. As for the negative examples ($w_{nv} = 0$), the comparison is not as straightforward, because the choice of terms in Eqs. 11 and 12 is not exactly the same. In particular, Eq. 11 holds $v'$ fixed and draws $v$ from the noise distribution, while Eq. 12 holds $v$ fixed and draws $v'$ from the noise distribution.

## C  DATA PREPROCESSING

In this paper we study Wikipedia articles (text8) and a corpus of U.S. Senate speeches. On both corpora, we restrict the vocabulary to the 15K most frequent words, replacing all the remaining words with a designated token. We annotate the data using NLTK tagger[9] or the Stanford CoreNLP tagger (Manning et al., 2014), using the universal tagset shown in Table 7.

The Senate speeches contain a lot of boilerplate repetitive language; for this reason, we tokenize around 350 frequent phrases, such as SENATOR FROM ALABAMA or UNITED STATES, considering the entire phrase an individual vocabulary term. We apply the POS tagger before this tokenization step, and then we assign the NOUN tag to all phrases.

We split the data into training (90%), testing (5%), and validation (5%) sets. We use the validation set to assess convergence, as explained in the main text. We subsample the frequent words following Mikolov et al. (2013b); i.e., each word $w_n$ in the training set is discarded with probability

$$\text{Prob}(w_n \text{ is discarded}) = 1 - \sqrt{\frac{t}{\text{frequency}(w_n)}}, \tag{14}$$

where frequency($w_n$) denotes the frequency of word $w_n$, and $t = 10^{-5}$.

For each method, we use $|\mathcal{S}_n| = 10$ negative samples on the Wikipedia articles and $|\mathcal{S}_n| = 20$ negative samples on the Senate speeches. Following Mikolov et al. (2013b), we draw the negative samples from the unigram distribution raised to the power of 0.75.

---

[9]See http://nltk.org

