# OpenReview forum: "Word2net: Deep Representations of Language"
_ICLR.cc/2018/Conference — Reject_

### Official Review · AnonReviewer3 · 2017-11-26
**A natural but questionable extension to word2vec**

**Rating:** 5
**Confidence:** 5

**Review:**

The paper extends SGNS as follows. In SGNS, each word x is associated with vectors a_x and r_x. Given a set of context words C, the model calculates the probability that the target word is x by a dot product between a_x and the average of {r_c: c in C}.  The paper generalizes this computation to an arbitrary network: now each word x is associated with some network N_x whose input is a set of context words C and the output is the aforementioned probability. This is essentially an architectural change: from a bag-of-words model to a (3-layer) feedforward model.

Another contribution of the paper is a new form of regularization by tying a subset of layers between different N_x. In particular, the paper considers incorporating POS tags by tying within each POS group. For instance, the parameters of the first layer are shared across all noun words. (This assumes that POS tags are given.)

While this is a natural extension to word2vec, the reviewer has some reservations about the execution of this work. Word embeddings are useful in large part because they can be used to initialize the parameters of a network. None of the chosen experiments shows this. Improvement in the log likelihood over SGNS is somewhat obvious because there are more parameters. The similarity between "words" now requires a selection of context vectors (7) which is awkward/arbitrary. The use of POS tags is not very compelling (though harmless). It's not necessary: contrary to the claim in the paper, word embeddings captures syntactic information if the context width is small and/or context information is provided. A more sensible experiment would be to actually plug in the entire pretrained word nets into an external model and see how much they help.

EDIT: It's usually the case that even if the number of parameters is the same, extra nonlinearity results in better data fitting (e.g., Berg-Kirkpatrick et al, 2010), it's still not unexpected.

All of this is closely addressed in the following prior work:

Learning to Embed Words in Context for Syntactic Tasks (Tu et al., 2017)

Quality: Natural but questionable extension, see above.

Clarity: Clear.

Originality: Acceptable, but a very similar idea of embedding contexts is presented in Tu et al. (2017) which is not cited.

Significance: Minor/moderate, see above.

---

> ### Author Response · Authors · 2018-01-05
> **Contributions of Word2Net**
>
> Reviewer 3 gives a good summary of the paper and we thank them for useful references to related work.
>
> >> Improvement in the log likelihood over SGNS is somewhat obvious because there are more parameters.
>
> In our experiments we also compare models with the *same* number of parameters, e.g., by making the word vectors longer (see Tables 2, 5, and 6, the number of parameters per word p/V is in the third column). Our experimental study shows that word2net outperforms existing methods both with the same context dimension (K) and the same number of parameters (p/V), especially when we use POS information.
>
> >> The similarity between "words" (7) is awkward/arbitrary.
>
> We use Equation (7) to compute functional similarities between networks. It captures the idea that similar networks map similar inputs to similar outputs. The reason we use the context vectors as input is because they span typical inputs to the networks; linear combinations of context vectors have been used as input to the networks during training.
>
> >> The use of POS tags is not necessary: contrary to the claim in the paper, word embeddings capture syntactic information if the context width is small and/or context information is provided.
>
> As Reviewer 3 notes correctly, when we train a word embedding method such as word2vec, the embeddings of some words might capture some syntactic information, especially when the context size is small. However, work by Andreas and Klein, 2014, shows that over the entire vocabulary, the embeddings do not encode much syntactic information. One contribution of our work is to develop a method that allows us to incorporate syntactic information into the task of learning semantic representations.
>
> >> A more sensible experiment would be to actually plug in the entire pretrained word nets into an external model and see how much they help.
>
> We agree that evaluating word2net on downstream tasks is an excellent idea for future work. We also suspect that there are applications where existing embedding methods are not applicable since they only learn vectors, and the asymmetric compositionality of neural networks is necessary to solve the task.
>
> >> This is closely addressed in the following prior work: Learning to Embed Words in Context for Syntactic Tasks (Tu et al., 2017)
>
> The workshop paper of Tu et al. is very interesting. However, it addresses the entirely different task of learning a separate embedding for each occurrence of a word (i.e., token embeddings). The token embeddings are then used as features for downstream tasks such as POS tagging.
> In contrast, our aim is to learn a representation for each word depending on its POS tag, which means we use the POS tags during training as additional information we want the embedding to capture.
>
> >> Originality: Acceptable, but a very similar idea of embedding contexts is presented in Tu et al. (2017).
>
> We will include the citation to this interesting workshop paper (thank you for pointing us to it), but we also think that the main contribution of our work of learning a neural network for each word rather than a vector is complementary to the ideas in Tu et al.

---

### Official Review · AnonReviewer2 · 2017-11-27
**Nice idea but weak experimental section**

**Rating:** 4
**Confidence:** 4

**Review:**

The paper presents a method to use non-linear combination of context vectors for learning vector representation of words. The main idea is to replace each word embedding by a neural network, which scores how likely is the current word given the context words. This also allowed them to use other context information (like POS tags) for word vector learning. I like the approach, although not being an expert in the area, cannot comment on whether there are existing approaches for similar objectives.

I think the experimental section is weak. Most work on word vectors are evaluated on several word similarity and analogy tasks (See the Glove paper).  However, this paper only reports numbers on the task of predicting next word.

Response to rebuttal:

I am still not confident about the evaluation. I feel word vectors should definitely be tested on similarity tasks (if not analogy). As a result, I am keeping my score the same.

---

> ### Author Response · Authors · 2018-01-05
> **Evaluation of Word2Net**
>
> We thank Reviewer 2 for the thoughtful comments.
>
> >> I like the approach, although not being an expert in the area, cannot comment on whether there are existing approaches for similar objectives.
>
> While there is existing work to learn a vector for each word or a Gaussian distribution for each word, this is the first work to learn a neural network for each word.
>
> >> Most work on word vectors are evaluated on several word similarity and analogy tasks. However, this paper only reports numbers on the task of predicting next word.
>
> Thank you for the comment. In unsupervised learning methods, held-out predictions are a standard evaluation of model fitness. Our evaluation shows that due to word2net's capacity to learn nonlinear relationships between contexts and word occurrence, it fits the data better than existing methods.
>
> The focus of our paper is not to obtain better analogies. While there are numerous papers on embeddings that are not evaluated on analogies, we agree that this is an interesting point to explore in future work.

---

### Official Review · AnonReviewer1 · 2017-11-28
**Another tweak on learning word embeddings**

**Rating:** 4
**Confidence:** 4

**Review:**

This paper presents another variant on neural language models used to learn word embeddings. In keeping with the formulation of Mikolov et al, the model learned is a set of independent binary classifiers, one per word. As opposed to other work, each classifier is not based on the dot product between an embedding vector and a context vector but instead is a per-word neural network which takes the context as input and produces a score for each term. An interesting consequence of using networks instead of vectors to parametrize the embeddings is that it's easy to see many ways to let the model use side information such as part-of-speech tags. The paper explores one such way, by sharing parameters across networks of all words which have the same POS tag (effectively having different parameterizations for words which occur with multiple POS tags).

The idea is interesting but the evaluation leaves doubts. Here are my main problems:
 1. The quantitative likelihood-based evaluation can easily be gamed by making all classifiers output numbers which are close to 1. This is because the model is not normalized, and no attempt at normalization is claimed to be made during the likelihood evaluation. This means it's likely hyperparameter tuning (of, say, how many negative examples to use per positive example) is likely to bias this evaluation to look more positive than it should.
 2. The qualitative similarity-based evaluation notes, correctly, that the standard metric of dot product / cosine between word embeddings does not work in the case of networks, and instead measures similarity by looking at the similarity of the predictions of the networks. Then all networks are ranked by similarity to a query network to make the now-standard similar word lists. While this approach is interesting, the baseline models were evaluated using the plain dot product. It's unclear whether this new evaluation methodology would have also produced nicer word lists for the baseline methods.

In the light that the evaluation has these two issues I do not recommend accepting this paper.

---

> ### Author Response · Authors · 2018-01-05
> **Learning word networks rather than word vectors**
>
> We thank Reviewer 1 for the excellent feedback. The summary shows a firm understanding of our work and we appreciate the constructive feedback for the experimental section.
>
> >> Like Mikolov et al, the model learned is a set of independent binary classifiers, one per word.
>
> Correct. One contribution of this work is to demonstrate that the skipgram formulation of Mikolov et al., originally derived as an approximation to multiclass classification, can be viewed as a binary classification, one for each word. The binary classifiers are not fully independent; rather, they are coupled through the context vectors, which are the input to the classifiers.
>
> >> An interesting consequence of using networks instead of vectors is that it's easy to use side information such as part-of-speech tags.
>
> Correct. Incorporating side information into embeddings is a difficult task and has only been partially addressed in previous work. In particular, syntactic information has been difficult to incorporate.
>
> >> The paper explores one such way, by sharing parameters across networks of all words which have the same POS tag.
>
> Yes, we effectively learn a different representation for each word - POS tag combination.
>
> >> The quantitative likelihood-based evaluation can easily be gamed.
>
> It is true that the quantitative evaluation can be gamed without proper normalization. To avoid that, we were careful to make the comparison as fair as possible. During training, all methods used the same number of negative samples, and the reported held-out likelihood values correspond to the average over both positive and negative samples, giving the same weights to positive and negative samples.
>
> >> In the qualitative similarity-based evaluation the vector representations are ranked by cosine similarity while the word networks are ranked by functional similarity (cosine similarity does not apply). While this approach is interesting, it is unclear if it would produce nicer lists for the baseline methods as well.
>
> Thank you. Evaluating the word vectors on "functional similarity" is an excellent idea. We reran the similarity queries and found that some queries changed slightly while others did not change. For example, in the 4 queries in Table 3, when evaluating the CBOW baseline with the functional similarity, only 2 queries changed by one word, while the other two queries remained unchanged. We leave it for future work to study the difference between functional and cosine similarity. If the functional similarity we developed for word2net is indeed better for vector-based embeddings this might be a worthwhile contribution by itself.

---

### Decision · Program_Chairs · 2018-01-29
**ICLR 2018 Conference Acceptance Decision**

**Decision:**

Reject

**Comment:**

Pros
-- Extends embeddings to use a richer representation; simple yet interesting improvement on Mikolov et al. work.
Cons
-- All of the reviewers pointed out that the experimental evaluations needs improvement. The authors should find better ways to improve both quantitative (e.g., accuracy in analogies as in Mikolov et al., or by using the model for an external task if that’s the end goal) and qualitative (using functional similarity for the baseline) evaluations.

Given these comments, the AC recommends that the paper be rejected.